# mRNA COVID-19 Vaccine Reactogenicity among Healthcare Workers: Results from an Active Survey in a Pediatric Hospital from Bucharest, January–February 2021

**DOI:** 10.3390/vaccines10060836

**Published:** 2022-05-25

**Authors:** Maria-Dorina Crăciun, Gabriela Viorela Nițescu, Mihaela Golumbeanu, Alina-Andreea Tănase, Daniela Pițigoi, Oana Săndulescu, Petru Crăciun, Bianca Georgiana Enciu, Radu Ninel Bălănescu, Alexandru Ulici

**Affiliations:** 1Department of Epidemiology, University of Medicine and Pharmacy “Carol Davila”, 020021 Bucharest, Romania; maria.craciun@umfcd.ro (M.-D.C.); daniela.pitigoi@umfcd.ro (D.P.); oana.sandulescu@umfcd.ro (O.S.); bianca.milcu@drd.umfcd.ro (B.G.E.); 2Emergency Clinical Hospital for Children “Grigore Alexandrescu”, 011743 Bucharest, Romania; mihaelagolumbeanu@yahoo.com (M.G.); radu.balanescu@umfcd.ro (R.N.B.); alexandru.ulici@umfcd.ro (A.U.); 3Department of Pediatrics, University of Medicine and Pharmacy “Carol Davila”, 020021 Bucharest, Romania; 4Emergency University Hospital, 050098 Bucharest, Romania; alina-andreea.tanase@rez.umfcd.ro; 5Department of Drug Industry, University of Medicine and Pharmacy “Carol Davila”, 020021 Bucharest, Romania; petru.craciun@drd.umfcd.ro; 6Department of Pediatric Surgery and Orthopedics, University of Medicine and Pharmacy “Carol Davila”, 020021 Bucharest, Romania

**Keywords:** active surveillance, COVID-19, mRNA vaccine, reactogenicity, healthcare workers

## Abstract

In Romania, health and social workers were prioritized for COVID-19 vaccination. We aimed to describe the vaccine adverse events identified through an active survey (using an electronic questionnaire) conducted among the staff of a pediatric hospital from Bucharest, vaccinated with the mRNA Pfizer-BioNTech vaccine. Data on the frequency and duration of adverse events were collected and analyzed using Microsoft Excel, Epi Info, and MedCalc. The questionnaire was sent to 426 persons. The participation rate was 81.2% after 1st dose and 63.8% after the 2nd dose. Overall, 81.9% were women, median age 42 (IQR 32–50 years). A total of 48 respondents (14.8%) reported no adverse event after the 1st dose and 35 (14.1) after the 2nd dose. No anaphylaxis was reported. The most frequent adverse event was pain at injection site, being reported by 261 responders (80.3%) after 1st dose and 187 (75.1%) after 2nd dose. Fatigue and headache were reported significantly less frequently in our study compared with data provided by the vaccine manufacturer. The current study has shown higher local reactogenicity after the first dose of the vaccine and higher systemic reactogenicity after the second dose. This real-world knowledge of the reactogenicity and safety profile may increase the vaccine’s acceptance rate among healthcare workers.

## 1. Introduction

The COVID-19 pandemic is one of the most significant challenges of the 21st century because of its morbidity, mortality, and significant economic and social impact.

Since the beginning of the pandemic, the efforts of physicians, researchers, pharmaceutical companies and regulatory agencies have focused on fast development and approval of safe and efficient vaccines. 

Thus, in December 2020, one year after the identification of the first cases of pneumonia of unknown etiology in Wuhan, China, the first vaccine against COVID-19 was granted conditional marketing authorization by the European Medicines Agency. This mRNA-based vaccine produced by Pfizer-BioNTech New York, United States of America, was initially approved by the US Food and Drug Administration. Shortly afterwards, other vaccines against COVID-19 also received approval [1,2,3].

The availability and access to COVID-19 vaccines were limited in the initial phases of the vaccination campaign, due to the insufficient production capacity and due to the very high demand. For this reason, the World Health Organization (WHO) and the European Centre for Disease Control and Prevention (ECDC) issued a series of recommendations in order to prioritize vaccination for certain risk groups. Thus, in Romania, as in other European countries, the vaccination was implemented in phases (1st phase—health and social services staff, 2nd phase—population at risk and those employed in the essential sectors, 3rd phase—general population) [4,5,6,7]. 

The COVID-19 vaccination campaign started in Romania on 27 December 2020, medical staff being the first category planned for vaccination. The vaccination process took place in the initial phases only in dedicated vaccination centers, and the vaccination was based on appointment [6,7]. The functional structure and the provision of the vaccination centers with equipment and staff is regulated by law [8]. 

Ascertaining the real time status of administered vaccinations and of vaccine adverse events is extremely useful for the proper implementation of any vaccination campaign or program, guiding each decision of the public health authorities [6,7,9].

Vaccine hesitancy is one of the main issues that public health authorities have faced, being the main cause of low vaccination coverage and the driver of epidemics of vaccine-preventable diseases. There is a general hesitancy in using a new vaccine, a vaccine which people consider that insufficient information exists in terms of safety and efficacy [9,10].

According to the Centre for Diseases Control, one of the most efficient methods through which the authorities can improve the confidence of the population in a vaccine is the transparent communication of information concerning its safety and efficacy [9]. 

In Romania adverse events following vaccination are passively recorded according to a surveillance methodology through filling-in the reporting form and an event investigation report (only for serious adverse events) to the National Centre for Surveillance and Control of Communicable Diseases. Furthermore, the pharmacovigilance system managed by the National Agency for Medicines might be used. Furthermore, there is a protocol for collaboration and exchange of information between the institutions supervising undesirable vaccine adverse events [11,12,13]. 

We aimed to describe the adverse events following COVID-19 immunization identified through an active survey conducted among healthcare workers of a Bucharest pediatric hospital, vaccinated during the first phase of the national vaccination campaign in Romania. 

## 2. Materials and Methods

An active survey of vaccine adverse events was performed in January and February 2021 and involved the hospital staff vaccinated against COVID-19 with the mRNA Pfizer-BioNTech vaccine with two doses of vaccine, 21 days apart. The two doses (1st and 2nd) were identical, 0.3 mL (30 mcg), as recommended by the vaccine manufacturer, and were administered in the same vaccination center.

The appointment for vaccination was organized by the hospital, through the online platform of the Ministry of Health, based on the lists of the staff having opted for vaccination. 

A specifically designed Google Forms questionnaire was sent by the Hospital Department of Infections Control together with the invitation to the survey to all the persons appointed to be vaccinated with their first doses in January. The contact details (either telephone number, e-mail address, or both) for the vaccination appointment list were used. The questionnaire was sent once for both doses at the beginning of the vaccination campaign.

The participants were asked to answer the questionnaire regarding the adverse events noticed in the first 3 days after the administration of each vaccine dose, without specifying a due date for submission. The time required for filling the questionnaire was estimated to be about 2 min.

The adverse events listed in the questionnaire were those of vaccine leaflet [14]. Additionally, an open question allowed participants to describe any other adverse events that might have occurred, independently from a potential pre-existing association with the vaccine. For simplicity, it was required that the adverse event was to be recorded as such, without details concerning its severity or duration. The questionnaire also elicited information referring to age, gender, history of chronic diseases or allergies, as well as possible medical examination or treatment for the adverse events that had occurred.

The survey protocol was approved by the Hospital Ethics Commission, with the registration number 1176/15 January 2021.

The adverse events data were collected on behalf of the hospital in a database managed by the Department of Infections Control, apart from the national database for adverse events [12,13]. For data analysis, Microsoft Excel, Epi Info version 7.2.2.2, and MedCalc for Windows version 20.4 applications were used (MedCalc Software, Ostend, Belgium) [15,16]. The questionnaires received were checked and those with recorded errors (e.g., date of birth instead of vaccination date), duplicates (identical questionnaires submitted twice), and received too early (before the 3rd day after the administration of the vaccine) were excluded.

## 3. Results

The vaccination started on 4 January 2021. The questionnaire was sent to 426 persons scheduled for vaccination during January–February. During the study period (18 January–25 February), 656 answers were collected (Figure 1).

The participation rate, calculated after having eliminated only duplicates, was 81.2% for the 1st dose and 63.8% for the 2nd dose.

The group of participants consisted mainly of women (81.9%), both for 1st dose (80.9%), and for 2nd dose (83.1%), corresponding to the gender structure of the hospital staff.

The median age was 42 (IQR 32–50 years); for the 1st dose the median age was 42 (IQR 31–49 years) and for the 2nd dose the median age was 43 (IQR 33–50 years). 

For comparison with the data of the vaccine manufacturer [17], the participants in the survey were analyzed into 2 age groups: 18–55 and over 55, the 18–55 age group representing 89.2% for the 1st dose and 90.4% for the 2nd dose (Table 1).

The median interval from vaccination and response was 8 days (IQR 5–14 days); for the 1st dose this was 9 days (IQR 5–15 days) and for the 2nd dose it was 7 days (IQR 4–13 days).

The analysis of vaccine adverse events was based on the valid questionnaires for each specific vaccine dose. Overall, 48 participants (14.8%) for the 1st dose and 35 participants (14.1%) for the 2nd dose did not report any vaccine adverse event. For the 2nd dose, 164 participants (65.9%) reported more than one adverse event, a statistically significant higher value than those for the 1st dose, 121 participants (37.2%).

The most frequent local adverse event was pain at the injection site, reported by 261 participants (80.3%) for the 1st dose and 187 participants (75.1%) after the 2nd dose. Redness was reported by 17 participants (5.2%) for the 1st dose and 13 participants (4.8%) after the 2nd dose. Swelling was reported by 26 participants (8.0%) for the 1st dose and 20 participants (5.6%) after the 2nd dose. There was no statistically significant difference between the frequency of the local adverse events reported for the 1st dose and for the 2nd dose.

The most frequent systemic adverse events were fatigue, reported by 60 participants (18.5%) for the 1st dose and 114 participants (45.8%) for the 2nd dose, headache, reported by 56 participants (17.2%) for the 1st dose and 85 participants (34.1%) for the 2nd dose, respectively, and muscle pain, reported by 39 participants (12.0%) for the 1st dose and 80 participants (32.1%) for the 2nd dose. The digestive symptoms were extremely rare (0 cases of diarrhea, 1 episode of vomiting). 

A statistically significant higher frequency of systemic adverse effects after the 2nd dose was observed for most adverse events: fever, fatigue, headache, chills, nausea, muscle pain, joint pain, insomnia, and lymphadenopathy (Figure 2).

No anaphylactic reaction was reported by any participant.

The data obtained based on the survey for the 18–55 years age group were compared to those of the vaccine manufacturer for each dose [1,17]. The over 55 years age group was not analyzed due to the small number of observations (10.3% of the participants). As the time interval between the vaccination and the response to the questionnaire was variable, the difference between the frequency of the adverse events reported within less than 7 days and those reported in more than 7 days, per doses, was analyzed. The 7-day limit was chosen for comparability with the data of the vaccine manufacturer, reported after the 7th day. No statistically significant difference was observed regarding the frequency of the reported adverse events (both for the 1st dose and for the 2nd dose) in connection with the time interval in which the questionnaire was answered. Therefore, the comparative analysis with vaccine manufacturer data was performed based on all valid responses. 

Comparing our results with the vaccine manufacturer data, there were no statistically significant differences with respect to the frequency of the local adverse events reported for both the 1st dose and the 2nd dose (Table 2 and Table 3). 

Regarding the frequency of the systemic adverse events, in the surveyed group a significantly lower frequency was observed for fever, headache, fatigue, chills, and myalgia after the 1st dose (Table 2). For the 2nd dose, events such as fatigue, headache, and chills were reported with a significantly lower frequency among the surveyed group (Table 3).

An extremely small number of participants (5; 0.9%) reported having required medical examination and treatment following the adverse events (2 after the 1st dose and 3 after the 2nd dose). A case of brachial paresis was reported after the 2nd dose. The paresis fully recovered under non-steroidal anti-inflammatory and cortisone treatment.

## 4. Discussion

Knowing the real-world incidence of vaccine adverse events is essential for a proper implementation of a vaccination campaign. Knowing and transparently communicating the information concerning the reactogenicity of a vaccine can increase the confidence in vaccines, which is particularly important for a newly-introduced vaccine [18]. A good method to observe vaccine adverse events is active surveillance. The major advantages of active surveillance are better knowledge (as close as possible to reality) of the frequency and type of adverse events that occur, as well as a higher quality of data. 

Although the need to know the real frequency of COVID-19 vaccine adverse events is high, the implementation of an active surveillance system concerning vaccine adverse events is difficult, due to the specific constraints, high cost, and higher human resources requirements.

Within the survey, our aim was to actively observe the real-world rate of occurrence of vaccine adverse events among the healthcare workers vaccinated during the first COVID-19 vaccination phase in the country.

During the survey, the Department of Infections Control provided feedback to the staff about the survey participation, with regards to the reported adverse events, through daily staff meetings. To avoid the risk of influencing future answers to the questionnaire, only general information was provided. Open communication was considered essential for the staff and a factor driving the vaccination rate increase.

The frequency of systemic adverse events was lower among our respondents when compared with those reported by the vaccine manufacturer, but in line with other studies (UK, Israel) [18,19,20].

This study has a set of limitations. First, a certain degree of recall bias cannot be excluded, although the timespan from vaccination to survey completion (median 8 days, IQR 5–14 days) was generally short enough to ensure reliability of the recalled data. Second, our study is limited by the fact that the questionnaire was not reapplied in September 2021, when the vaccine booster dose started to be administered, and thus we were unable to assess the reactogenicity of the 3rd vaccine dose. This was not possible due to the overall low uptake of the booster dose in Romania (8.8% according to the data reported to the European Centre for Disease Control on 7 April 2022) [21]. However, since the focus of the current survey was to compare real-world data with manufacturer reported data from pivotal clinical trials, the lack of information for the booster dose does not hinder the findings from the current analysis in any way. Other limitations of the survey, which was carried out based on anonymous questionnaires, were the existence of duplicate answers and errors, which were managed during the validation phase, as well as a lower response rate after the administration of the 2nd dose, potentially caused by the transmission of the questionnaire once for both doses. 

The strengths of this survey include the high quality of the data, the healthcare workers being able to recognize and describe the symptoms observed easily and precisely. At the same time, the data were collected immediately after the start of the vaccination campaign using a new vaccine, obtained through a novel technology, and the concern of the healthcare workers in terms of reactogenicity has been important, as indicated by the high rate of participation (over 80%) for the 1st dose. The lower response rate for the 2nd dose can be explained by a drop of interest, as well as by the long-time interval between the receipt of the questionnaire and the moment of vaccination (the questionnaire was sent only once for both doses). 

The 3-day interval for the observation of the adverse events was chosen based on the observation in the survey of the vaccine manufacturer, which showed that most systemic adverse events were reported (both for the 1st dose and for the 2nd dose) in the first two days after vaccination, and persisted for one more day.

The high share of women in the surveyed group reflects the gender distribution among the hospital staff. The results reported in the survey refer to the adult population (healthcare workers).

## 5. Conclusions

The active surveillance study has shown the highest frequency of the adverse events for pain at the injection site. Fatigue and headache followed, but at a much lower level. The digestive symptoms were extremely rare.

There was no difference between the frequency of the local adverse events reported for the 1st dose and the 2nd dose. A higher frequency of systemic adverse events after the 2nd dose was observed for most reactions: fever, fatigue, headache, chills, nausea, muscle pain, joint pain, insomnia, and lymphadenopathy. There were no anaphylactic reactions or other severe adverse events requiring hospitalization or causing after-effects or invalidity.

The reactogenicity among the healthcare workers was similar to that specified in the summary of the product characteristics in the same age group for local adverse events. For some systemic adverse events it was lower, for both doses. The higher systemic adverse events frequency after the administration of the 2nd dose is consistent with the vaccine manufacturer data. 

The active surveillance provides useful information concerning the post-authorization safety of the vaccines administered in real-world settings.

The data collection through the online questionnaire method allowed real-time monitoring of the vaccine adverse events. This method of online questionnaire was brief and straightforward for the responders, particularly important when surveying healthcare workers during the COVID-19 pandemic, leading to a good uptake.

## Figures and Tables

**Figure 1 vaccines-10-00836-f001:**
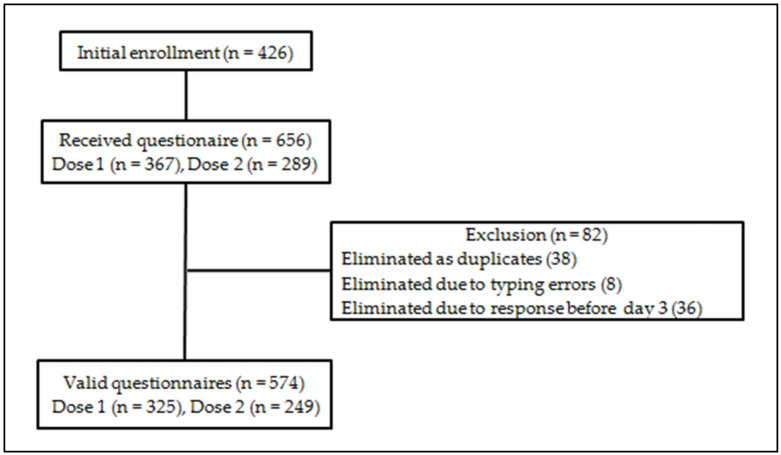
Flow chart of data exclusion for analysis, hospital survey, Bucharest, Romania, January–February 2021.

**Figure 2 vaccines-10-00836-f002:**
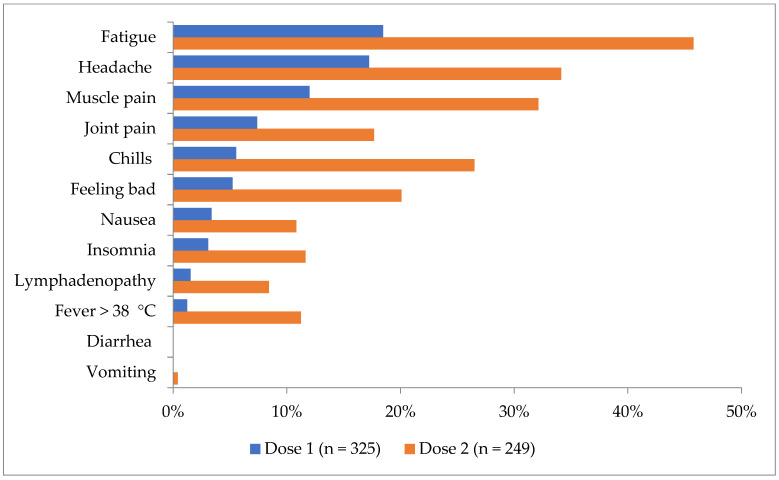
Frequency of systemic adverse events by dose rank, hospital survey, January–February 2021.

**Table 1 vaccines-10-00836-t001:** Characteristics of participants by dose rank, hospital * survey, January–February 2021.

Characteristics		Dose 1 (n = 325)	Dose 2 (n = 249)	Total (n = 574)
Median age (years), IQR		42, IQR 31–49	43, IQR 33–50	42, IQR 32–50
Age group	18–55 years of age	290 (89.2%)	225 (90.4%)	515 (89.7%)
	56 years of age and older	35 (10.8%)	24 (9.6%)	59 (10.3%)
Gender distribution	Female	263 (80.9%)	207 (83.1%)	470 (81.9%)
	Male	62 (19.1%)	42 (16.9%)	104 (18.1%)

* “Grigore Alexandrescu” Emergency Clinical Hospital for Children, Bucharest, Romania.

**Table 2 vaccines-10-00836-t002:** Frequency of local and systemic adverse events after the 1st vaccine dose for the 18–55 years age group, hospital survey, January–February 2021.

Adverse Event	Hospital Survey (n = 290) Number (%)	Reference Study- Pfizer-BioNTec [1] (n = 2291) Number (%)	Statistical Analysis *
Local reactions			
Redness (any)	16 (5.5)	104 (4.5)	*p* = 0.44, χ^2^(1) = 0.58
Swelling (any)	25 (8.6)	132 (5.8)	*p* = 0.06, χ^2^(1) = 3.51
Pain at the injection site (any)	244 (84.1)	1904 (83.1)	*p* = 0.67, χ^2^(1) = 0.18
Systemic reactions			
Fever > 38 °C	3 (1.0)	85 (3.7)	*p* = 0.02, χ^2^(1) = 5.71
Fatigue (any)	55 (19.0)	1085 (47.4)	*p* < 0.0001, χ^2^(1) = 84.14
Headache (any)	52 (17.9)	959 (41.9)	*p* < 0.0001, χ^2^(1) = 62.18
Chills (any)	14 (4.8)	321 (14.0)	*p* < 0.0001, χ^2^(1) = 19.29
Vomiting (any)	0	28 (1.2)	*p* = 0.06, χ^2^(1) = 3.51
Muscle pain (any)	35 (12.1)	487 (21.3)	*p* < 0.0001, χ^2^(1) = 13.47
Joint pain (any)	22 (7.6)	251 (11.0)	*p* = 0.08, χ^2^(1) = 3.14

* N−1 chi-squared test for the comparison of two proportions (from independent samples).

**Table 3 vaccines-10-00836-t003:** Frequency of local and systemic adverse events after the 2nd vaccine dose for the 18–55 years age group, hospital survey, January–February 2021.

Adverse Event	Hospital Survey (n = 225) Number (%)	Reference Study- Pfizer-BioNTech [1] (n = 2098) Number (%)	Statistical Analysis *
Local reactions			
Redness (any)	11 (4.9)	123 (5.9)	*p* = 0.54, χ^2^(1) = 0.37
Swelling (any)	13 (5.8)	132 (6.3)	*p* = 0.76, χ^2^(1) = 0.09
Pain at the injection site (any)	176 (78.2)	1632 (77.8)	*p* = 0.89, χ^2^(1) = 0.01
Systemic reactions			
Fever > 38 °C	26 (11.6)	331 (15.8)	*p* = 0.10, χ^2^(1) = 2.75
Fatigue (any)	107 (47.6)	1247 (59.4)	*p* = 0.0006, χ^2^(1) = 11.63
Headache (any)	81 (36.0)	1085 (51.7)	*p* < 0.0001, χ^2^(1) = 20.02
Chills (any)	60 (26.7)	737 (35.1)	*p* = 0.011, χ^2^(1) = 6.36
Vomiting (any)	1 (0.4)	40 (1.9)	*p* = 0.10, χ^2^(1) = 2.65
Muscle pain (any)	75 (33.3)	783 (37.3)	*p* = 0.24, χ^2^(1) = 1.39
Joint pain (any)	40 (17.8)	459 (21.9)	*p* = 0.15, χ^2^(1) = 2.02

* N−1 Chi-squared test for the comparison of two proportions (from independent samples).

## Data Availability

The datasets generated and analyzed during the current study are available from the corresponding author upon reasonable request.

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
