# Peer review of "mRNA COVID-19 Vaccine Reactogenicity among Healthcare Workers: Results from an Active Survey in a Pediatric Hospital from Bucharest, January–February 2021"

_vaccines, 2022, doi:10.3390/vaccines10060836_

Round 1

Reviewer 1 Report

 In this manuscript the authors describe the COVID-19 vaccine adverse events identified through an active survey (using an electronic questionnaire) conducted among the staff of a pediatric hospital from Bucharest, vaccinated with mRNA Pfizer-BioNTech vaccine during the first phase of the national vaccination campaign in Romania.

In this contribution, the participants were asked to complete a questionnaire describing the reactions observed in the first 3 days after the administration of each dose of vaccine. The questionnaire also elicited information referring to age, gender, history of chronic diseases or allergies, as well as possible medical examination or treatment for the adverse events that had occurred..

The authors established that there was no statistically significant difference between the frequency of the local reactions reported for the 1st dose and for the 2nd dose. However,  examination of the frequency of systemic reaction, by dose rank showed a statistically significant higher frequency of systemic adverse effects after the 2nd dose was observed for most reactions: fever, fatigue, headache, chills, nausea, muscle pain, joint pain, insomnia and lymphadenopathy.

In conclusion, the authors found that the reactogenicity among the healthcare workers reported during the active survey was similar with that specified in the summary of the product characteristics in the  same age group for local reactions. For some systemic reactions it was lower, for both  doses. The systemic reactions were more frequent after the administration of the 2nd dose, an observation consistent with the vaccine manufacturer data.

This manuscript presents an interesting active survey application to monitor whether there was statistically significant difference between the frequency of the local reactions reported for the 1st dose and for the 2nd dose of the COVID-19 vaccine. This manuscript is well prepared, with current references and the findings are very interesting. The introduction and approach to sample preparation are thorough.

However, the authors have indicated that their study has a set of limitations. First, a certain degree of recall bias cannot be excluded, although the timespan from vaccination to survey completion was generally short enough to ensure reliability of the recalled data.  Second, this study is limited by the fact that the questionnaire was not reapplied in September 2021, when the booster dose of vaccine started to be administered, and thus the authors were unable to assess the reactogenicity of the 3rd vaccine

I have some minor comments that needs to be answered:

  1. Please correct paediatric for pediatrics.

  1. When you discuss of first and second doses of the COVID-19 vaccine are we safe to assume that the dosage of the mRNA Pfizer-BioNTech vaccine are identical?

This is not clear in your manuscript. And if you search in the literature you will find that the COMIRNATY multiple dose vial with a purple cap and purple label border contains a volume of 0.45 mL. It is supplied as a frozen suspension that does not contain preservative. Each vial must be thawed and diluted prior to administration. Vial Cap and Label Border Colour: Purple; Grey and Orange. So, the question here are the two doses (1st and 2nd) identical. Please clarify.

Author Response

Thank you for the revision of our manuscript.

We include a point-by-point response to the Reviewers yours comments below.

Reviewer 2 Report

Teh authors report adverse events regarding mRNA COVID19 vaccine.

Despite the idea could be interesting just for the suvery sent to the vaccinated subjects there are some flawless that should be corrected

Major Criticism

How patients were selected? it is not clear the selection strategy

Are they reppresentative of a single Romania districit/region/city?

The suvery was sent after how many dys from vaccination? the AE are reported according to time elapsed form vaccination or just as onetime evaluation without further details?

The authors stated that the data were collected in national AE registry, please provide a correct classification of AE and not in the present form as it was reffered to the safety evaluation of a clinical trial. This report should be related to AE and therefore could be stratificated according to standard classification of AE

Minor Criticism

Adjust figure and tables providing more details about strategy and localization of the patients

Improve conclusion

Adbridge introduction

Author Response

Thank you for the revision of our manuscript.

We include a point-by-point response to your comments below.

Round 2

Reviewer 2 Report

The current version has been improved.

However before acceptance please specify, in materil and method, that your AE were not collected in National registry for AE and that they have been evaluated and collected in a hospital database

Author Response

Thank you very much for your comments regarding methodology.

We have improved the section as recommended.